# A Deep Learning Approach for Detecting Stroke from Brain CT Images Using OzNet

**DOI:** 10.3390/bioengineering9120783

**Published:** 2022-12-08

**Authors:** Oznur Ozaltin, Orhan Coskun, Ozgur Yeniay, Abdulhamit Subasi

**Affiliations:** 1Institute of Science, Department of Statistics, Beytepe Campus, Hacettepe University, Ankara 06800, Turkey; 2Gaziosmanpasa Training and Research Hospital, Pediatric Neurology, Health Sciences University, Gaziosmanpasa, Istanbul 34779, Turkey; 3Institute of Biomedicine, Faculty of Medicine, University of Turku, 20520 Turku, Finland; 4Department of Computer Science, College of Engineering, Effat University, Jeddah 21478, Saudi Arabia

**Keywords:** brain stroke, classification, convolution neural networks, computed tomography, feature extraction, mRMR, OzNet

## Abstract

A brain stroke is a life-threatening medical disorder caused by the inadequate blood supply to the brain. After the stroke, the damaged area of the brain will not operate normally. As a result, early detection is crucial for more effective therapy. Computed tomography (CT) images supply a rapid diagnosis of brain stroke. However, while doctors are analyzing each brain CT image, time is running fast. This circumstance may lead to result in a delay in treatment and making errors. Therefore, we targeted the utilization of an efficient artificial intelligence algorithm in stroke detection. In this paper, we designed hybrid algorithms that include a new convolution neural networks (CNN) architecture called OzNet and various machine learning algorithms for binary classification of real brain stroke CT images. When we classified the dataset with OzNet, we acquired successful performance. However, for this target, we combined it with a minimum Redundancy Maximum Relevance (mRMR) method and Decision Tree (DT), k-Nearest Neighbors (kNN), Linear Discriminant Analysis (LDA), Naïve Bayes (NB), and Support Vector Machines (SVM). In addition, 4096 significant features were obtained from the fully connected layer of OzNet, and we reduced the dimension of features from 4096 to 250 using the mRMR method. Finally, we utilized these machine learning algorithms to classify important features. As a result, OzNet-mRMR-NB was an excellent hybrid algorithm and achieved an accuracy of 98.42% and AUC of 0.99 to detect stroke from brain CT images.

## 1. Introduction

A stroke is the most general neurological reason for death and inability worldwide [1,2]. A stroke, also specifying a cerebrovascular injury, occurs as a result of the brain arteries’ ischemia or hemorrhage and commonly causes diverse motor and cognitive damages that risk functionality [3]. Approximately 16 million people suffer from stroke in the world [3]. The primary stage is the early detection of the stroke. Notably, when determining the cause of injury made to the brain cells, the doctors significantly benefit from brain imaging techniques. CT and Magnetic resonance imaging (MRI) are the imaging techniques for brain strokes. When diagnosing the stroke, an MRI is generally used. However, in emergencies, it is more advantageous to utilize a CT due to the time constraint. Therefore, a much more rapid assessment is vital for patients. The researchers targeted the rapid recognition of the disease from MRI or CT using artificial intelligence such as deep learning algorithms to help doctors [4,5,6].

In this paper, we proposed a convolution neural networks (CNN) architecture to classify brain stroke CT images effectively, called OzNet [7]. Here, we combined it with machine learning methods such as DT, kNN, LDA, NB, and SVM. Primarily, OzNet has been used for a binary (stroke and normal) classification of the CT dataset. Although OzNet obtained acceptable results, we utilized it for a deep feature extraction from the images. In this stage, 4096 features were obtained from OzNet’s fully connected layer. Then, we used a minimum Redundancy Maximum Relevance (mRMR) method to reduce the dimension of the features, and 250 features were selected with this method. Thus, we classified important features using machine learning algorithms. Figure 1 shows several hemorrhagic stroke CT images, and arrows display the stroke lesion. In this paper, we utilized these images in the "stroke" class. As a result, the main aim of this paper is to present the best structure for detecting stroke from brain CT images. Our hypothesis is that OzNet gives better results than previous works.

The rest of the paper is arranged as follows:

We presented literature review in Section 2. Then, we briefly represented the dataset and methods in Section 3. We interpreted the performance metrics for each experiment in Section 4. We also discussed the results and compared them with prior studies in Section 4. The conclusion is given in Section 5.

## 2. Literature Review

Jayachitra and Prasanth [8] proposed a new optimized fuzzy level segmentation algorithm to determine the stroke lesions. Then, they extracted the multi-textural features to compose a feature set. In addition, they classified these features with the proposed weighted Gaussian Naïve Bayes as normal and abnormal (stroke) classes. As a result, they obtained a 99.32% accuracy, 96.87% sensitivity, and 98.82% F1 measure using the proposed method.

Subuddhi et al. [9] used an MRI, which is generally utilized for the correct diagnosis of stroke. Essentially, they introduced an algorithm having the decision system to ascertain the stroke utilizing MRI images’ diffusion-weighted image sequence.

Additionally, their study included both segmentation and classification parts. Primarily, they expressed that the stroke has three classes: partial anterior circulation syndrome, lacunar syndrome, and total anterior circulation stroke. Next, they segmented the region of the stroke by applying an expectation–maximization algorithm. Further, in order to increase the detection accuracy, they utilized the fractional-order Darwinian particle swarm optimization technique. In the classification part, they utilized SVM and random forest (RF) classifiers for extracted features from segmented regions. Finally, they obtained an accuracy of 93.4% with the RF classifier.

Bento et al. [10] proposed an SVM for automatically detecting stroke from brain MRI. In addition, they possessed 401 samples with four classes and finally acquired an accuracy rate of 97.5%, a sensitivity of 96.4%, and a specificity of 97.9%.

Kasabov et al. [11] suggested a novel evolving spiking neural network reservoir system to predict cases and individualized modeling of Spectro-temporal data. When they compared their proposed method with traditional machine learning algorithms, such as multiple linear regression, multi-layer perceptron (MLP), and SVM, their experimental results showed that the proposed method had the highest accuracy of 94%.

Karthik et al. [12] used a deep fully convolutional network with the supervised approach for segmentation ischemic region. Furthermore, they highlighted the implementation of Leaky Rectified Linear Unit activation inside the last two layers of the architecture. In addition, they expressed that performing this method could learn extra features not being in U-Net architecture. As a result, they obtain a Dice coefficient of 0.70.

Rebouças Filho et al. [13] proposed a novel technique to extract features that propped up radiological density patterns of the brain and named Analysis of Brain Tissue Density. Moreover, they utilized this technique for extracting features from brain CT images. Additionally, to evaluate their proposed method, they utilized five machine learning algorithms: MLP, SVM, k-Nearest Neighbors (kNN), and the Optimum Path Forest (OPF). Consequently, they achieved the best accuracy of 99.30% with ABTD-OPF.

Vargas et al. [14] performed a classification with artificial neural networks to CT perfusion images using k-fold cross-validation. Further, they utilized 396 perfusion images and obtained an accuracy of 85.8%.

Dourado Jr. et al. [15] developed an IoT system to detect and classify stroke from brain CT images online. Additionally, in the extraction features phase, they used the pre-trained architectures DenseNet121, DenseNet169, DenseNet201, InceptionResNetv2, InceptionV3, MobileNet, NasNetLarge, NasNetMobile, ResNet50, VGG-16, VGG-19, and Xception to extract features from two types of brain images. Moreover, in the classification phase, they merged these with Machine Learning (ML) algorithms: Bayes classifier, MLP, kNN, RF, SVM (linear), and SVM (radial basis function). Generally, they stated that the experiments achieved very good results. However, CNN- kNN gave an accuracy of 100%, especially for both types of images. Notably highlighted in the study, the MobileNet-ML was the structure that yields the fastest results in terms of time consumption.

Li et al. [16] classified stroke-associated pneumonia data which is collected from the National Advanced Stroke Center of Nanjing First Hospital (China) including 3160 patients. In the pre-processing stage, they split into the data a training set and a testing set. Next, they classified the data with five ML algorithms: logistic regression, SVM, RF classifier, extreme gradient boosting (XGBoost), and fully connected deep neural network (DNN). In the experimental results, while they obtained the highest accuracy of 76.3% using DNN, they acquired the highest area under the curve (AUC) value of 0.841 utilizing XGBoost.

Gautam and Raman [17] collected brain CT images data from the Himalayan Institute of Medical Sciences, Dehradun, India. In the study, they suggested CNN architectures in order to classify brain CT images, which were included in three classes. In addition, they implemented 10-fold cross-validation, divided it into testing and training sets, and created two datasets: dataset 1, which included binary classes (hemorrhagic, ischemic), and dataset 2, which had three classes (hemorrhagic, ischemic, and normal). When they split into dataset 1 as a training set of 80% and a testing set of 20%, they acquired an accuracy of 98.33%. In addition, while they implemented tenfold cross-validation to dataset 1, they obtained an accuracy of 98.77%. When similar processes were carried out in dataset 2, they got an accuracy of 92.22% and 93.33%, respectively.

Bacchi et al. [18] studied clinical brain CT data and predicted the National Institutes of Health Stroke Scale of ≥4 scores at 24 h or modified Rankin Scale 0–1 at 90 days (“mRS90”) using CNN+ Artificial Neural Network hybrid structure. As a result, they acquired the best prediction of mRS90 an accuracy of 74% using the structure.

Saritha et al. [19] integrated wavelet entropy-based spider web plots and probabilistic neural networks to classify brain MRI, which were normal brain, stroke, degenerative disease, infectious disease, and brain tumor in their study. First, in the pre-processing stage, they used two dimensional (2D) discrete wavelet transform (DWT) for brain images. In the feature extraction stage, they used spider web plots, and in the classification stage, implemented a probabilistic neural network. In the final, they expressed that classification accuracy was attained 100%.

El-Dahshan et al. [20] used DWT to extract features from brain MRI images. Then, they diminished to these for obtaining more efficient features by utilizing principal component analysis. Next, the extracted features were classified as normal and abnormal cases by utilizing a feed-forward back-propagation artificial neural network and kNN. In their study, classification results were revealed that the best classifier was kNN with an accuracy of 98%.

Xu et al. [21] proposed a novel diagnostic tool for the health of the brain. Their study included two phases: classification and segmentation of brain stroke CT images. Currently, Deep Learning algorithms (DL), ML algorithms, and created hybrid algorithms with DL–ML approaches are utilized in many studies for detecting brain stroke [8,22,23,24,25,26,27,28,29,30,31].

When the studies in the literature were examined, it is seen that their performances are not successful (accuracy below 95%) in stroke detection. Hence, a new deep learning architecture, OzNet is developed to achieve better performance in stroke detection. This architecture is not only considered as a classification algorithm but also as a deep feature extractor from images automatically. Additionally, our suggested framework was not computationally complex when compared to other methods, such as the transfer learning methods and the performance of the developed framework better than the previous studies.

## 3. Materials and Methods

### 3.1. Dataset

The dataset was collected from Lady Reading Hospital, Peshawar, Khyber Pakhtunkhwa, Pakistan in 2012 [32]. Then, Afridi et al. [32] created a descriptive study to detect risk factors such as age, gender, smoking, diabetes mellitus, and hypertension for brain stroke. In total, they studied 100 patients whose ages were 16 years and above and whose gender rates were 68% males and 32% females. Additionally, the brain CT images of these patients include 1551 normal and 950 stroke classes and a size of 650 × 650 grayscale for each image. However, we randomly equalized the dataset in order to overcome overfitting while training. Therefore, a new dataset was composed of 950 normal and 950 stroke classes and also resized 227 × 227 for each image.

### 3.2. A Novel CNN Architecture

Nowadays, it has become very significant to get rapid and correct results, especially in the field of health. For this reason, deep learning algorithms are widely used by researchers. In this paper, we create a novel deep learning approach called OzNet for 2D images. Although OzNet is seen as an ordinary CNN, it is composed of special parameters, filter sizes, filter numbers, padding, stride, and layers for taking robust results on biomedical images [7], detailed in Table 1.

OzNet is designed as a novel CNN architecture composed of 34 layers. There are seven blocks, and each block consists of a convolutional layer, a maximum pooling layer, a ReLU (Rectified Linear Unit) activation function, and a batch normalization layer. Next, 2 fully connected layers, a dropout layer, a SoftMax layer, and a classification layer are linked at the end of blocks, respectively. In this paper, we choose the ReLU function because of faster than others [33].

When OzNet is performed as a classifier, a cross-entropy approach is utilized. In addition, we used it for feature extraction from the images owing to having many convolution layers, effectively.

In this paper, we compared OzNet with GoogleNet [34], Inceptionv3 [35], and MobileNetv2 [36] for detecting stroke from the brain CT images and applied 10-fold cross-validation for these architectures. Moreover, we used data augmentation on the brain stroke CT images dataset. Additionally, we selected stochastic gradient descent momentum (sgdm) as the optimization method, the momentum parameter as 0.95, and the learning rate as 0.0001.

### 3.3. Minimum Redundancy Maximum Relevance (mRMR) Method

Minimum Redundancy Maximum Relevance, (mRMR) is a feature selection method that tries to minimize the residual between the features while selecting the features with the highest amount of correlation with the class [37]. The level of association is usually characterized by either the correlation coefficient or the mutual information [38]. According to this approach, the m features with the highest correlation may not always be the best. The residuals of m number of features need to be examined. At the same time, features that minimize these residuals should be selected. This method is used for the reduction of features [39], and we also benefitted from it.

### 3.4. Decision Tree (DT)

The decision tree (DT), which is a machine learning algorithm, is performed for regression or classification [40,41]. Ross Quinlan created the program C4.5 over 20 years ago [42]. If any decision tree is implemented as a classification algorithm, it will possess a hierarchical structure.

A decision tree is a form for stating mappings. A tree is either a leaf node tagged with a class or the construction of a test node connected to two or more subtrees [43]. A test node calculates some results based upon the feature values of a sample, where each possible result is linked with one of the subtrees. A sample is classified by beginning at the stem node of the tree. If this node is a test, the result of the sample is defined, and the process maintains using the proper subtree. When a leaf is met in the end, its label offers the predicted class of the sample, and a decision tree can be built from a set of samples by a divide-and-rule strategy. If all the samples are interested in the same class, the tree is a leaf with that class as a label. Otherwise, a test is chosen that has different results for at least two of the samples that are divided according to this result. The tree has as its root a node determining the test, and for each result in turn, the corresponding subtree is acquired by implementing the same process to the subset of samples with that result [44].

### 3.5. k-Nearest Neighbor (kNN)

The k-Nearest Neighbors (kNN) algorithm is a supervised learning algorithm, and kNN is utilized for classification problems [45]. The state of nearest is calculated using a distance metric such as a Euclidean distance [42]. Notably, the essential component of the nearest neighbor algorithm assigns an input sample vector y, which is not a known classification, to its nearest neighbor class [46,47]. This idea can be widened to the k-nearest neighbor with the vector happening assigned to the class that is symbolized by a larger number among the k-nearest neighbors [46]. When thinking of more than one neighbor, there is a probability of an association among the classes with the largest number of neighbors in the k-nearest neighbor group. An easy way to resolve this issue is to restrict the k values [46]. In this study, the k value is selected according to the minimum error rate. Generally, increasing the k value improves the training number [48]. In this study, we use Euclidean distance as the distance metric, and we also identify the same distance weight and the k value as 10.

### 3.6. Linear Discriminant Analysis (LDA)

Ronald Fisher put forward the Linear Discriminant Analysis (LDA) in 1936 [49]. LDA is a widely performed structure for data classification and dimensionality reduction [50]. It happens in the identification of the projection hyperplane that decreases minimum into class variance and rises maximum the distance among the projected means of classes [49]. The two goals could be solved by tackling an eigenvalue problem relating to eigenvectors determining the hyperplane of regard [49]. Essentially, this situation is the same as Principal Component Analysis [50]. While it is explaining the importance of the granted features and classifying and reducing dimensions, this hyperplane can be applied [49].

### 3.7. Naïve Bayes (NB)

The Naive Bayes (NB) is one of the machine learning algorithms, and its structure is more flexible in handling a diversity of features [51]. Conditional distribution $p\left(\left.x_{i}\right|y\right)$ can be chosen according to the distribution of a feature $x_{i}$ [40]. If a feature is binary, non-binary discrete, and continuous, it can respectively be selected Bernoulli, multinomial, and Gaussian distribution. The number of parameters is linear with the number of features [7,40]. Learning algorithms and inference can be used with some closed-form solutions that are also linear in the number of various features [7,40]. As a result, this algorithm is scalable to main situations that contain a wide number of different features [7,40].

### 3.8. Support Vector Machine (SVM)

Support Vector Machine (SVM) is performed for regression, classification, and outlier analysis. Additionally, it is a well-known machine learning algorithm and is preferred by researchers. Cortes and Vapnik [52] presented this algorithm in 1995. Notably, SVM has many distinguishing properties. It can analyze many high-dimensional datasets and obtain a high accuracy rate for classification [53]. Additionally, it uses a kernel-based method for the effective division of the datasets for classification [54]. Furthermore, it has a flexible and suitable structure, so it can be utilized with many algorithms.

## 4. Results and Discussion

In this paper, we benefited from all presented classifiers. Notably, we performed OzNet for feature extraction, then we classified it with these algorithms. Thus, we designed the best structure for the classification of brain stroke CT images. Figure 2 demonstrates the flowchart of the paper.

Cross-validation is a confident method for classification. The dataset is split randomly with the determined number of folds, and considering one of the sub-folds as the test fold, it trains the framework with left behind folds [7,54]. The progress is iterated up to a number of folds and is tested in the framework [55]. In this paper, we implemented 10-fold cross-validation.

### 4.1. Performance Metrics

In this paper, we assess classifiers with performance metrics: accuracy, sensitivity, specificity, precision, F1-Score, and G-mean [56,57,58]. Here, TP: True Positive, FP: False Positive, TN: True Negative, and FN: False Negative are presented in Table 2.

### 4.2. Receiver Operating Characteristic (ROC) Curve

When any classifier’s performances are evaluated, the receiver operating characteristic (ROC) curve is generally utilized in problems of classification. This curve’s *x*-axis and *y*-axis indicate the false positive rate and true positive rate, respectively. In general, the AUC is also calculated to determine whether a particular condition exists regarding test data. If the AUC value is approximately 1, the classifier possesses very good performance [57,59]. In this paper, we also calculated AUC values for each classifier, and we demonstrated ROC curves.

### 4.3. Experimental Results

In this paper, we presented a novel CNN architecture, named OzNet, and utilized it to classify brain stroke CT images as stroke and normal classes. This architecture is one of the highlights of this paper. Moreover, we merged it with the mRMR method and the machine learning algorithms DT, kNN, LDA, NB, and SVM, and therefore, we designed new hybrid algorithms to find the best algorithm for brain stroke CT images classification. These hybrid algorithms are also other highlights of this paper. When we utilized these hybrid algorithms for classification, OzNet was tasked as a feature extractor from the images. This paper’s results were obtained in MATLAB via NVIDIA GeForce GTX 950M, 16 GB RAM, Intel Core i7-7500U CPU, 64-bit Operating System. First, we resized all images in 227 × 227 .jpg format in the preprocessing stage. Next, we classified brain stroke CT images using OzNet and compared it with GoogleNet [34], Inceptionv3 [35], and MobileNetv2 [36] for the same dataset. Further, we applied 10-fold cross-validation, which is a very reliable method for each classification [7,54]. The performance results are exhibited in Table 3.

According to Table 3, when the brain stroke CT images were classified with Goog-leNet, Inceptionv3, MobileNetv2, and OzNet, it was seen that the best architecture is OzNet with an accuracy of 87.19% and 0.9488 AUC value. Next, the better one is MobileNetv2 with an accuracy of 87.36%, and GoogleNet and Inceptionv3 have accuracies of 79.42% and 75.74%, respectively.

Though the performance of OzNet is acceptable, we combine it with classical ML algorithms. First, we trained OzNet and saved this network. Then, we split the dataset again training 70% and testing 30%, and hence, activated this network. Eventually, 4096 features were obtained from FC-8 (fully connected layer name, shown in Table 1) for each image. Then, we classified these features with classical ML algorithms. Table 4 displays performance metrics of the hybrid algorithms.

When we examined the performance metrics of OzNet in Table 4, essentially it was seen as successful with an accuracy of 87.87% and similar to other performance metrics.

According to Table 4, we can express that the highest performance belongs to the OzNet-SVM hybrid algorithm with an accuracy of 96.54%. It also has a sensitivity of 95.94%, a specificity of 97.14%, a precision of 97.11%, an F1 score of 96.54%, and an AUC value of 0.9921.

When we investigated in Table 4, OzNet-DT obtained an accuracy of 90.38% and sensitivity, specificity, precision, F1-score, G-mean, and AUC value of 90.23%, 90.53%, 90.5%,90.36%, 90.38%, and 0.9238, respectively.

When we viewed the results in Table 4, OzNet-kNN achieved an accuracy of 95.34% and sensitivity, specificity, precision, F1-score, G-mean, and AUC value of 94.89%, 95.79%, 95.75%,95.32%, 95.34%, and 0.9534, respectively.

When we analyzed in Table 4, OzNet-LDA achieved an accuracy of 96.39% and sensitivity, specificity, precision, F1-score, G-mean, and AUC value of 96.69%, 96.09%, 96.11%,96.40%, 96.39%, and 0.9937, respectively. The hybrid algorithm was also successful, and the AUC value was close to 1. However, its accuracy was lower than OzNet-SVM. Therefore, we declared that the best one is OzNet-SVM in this stage.

When we analyzed in Table 4, OzNet-NB had an accuracy of 96.32% and sensitivity, specificity, precision, F1-score, G-mean, and AUC value of 97.74%, 94.89%, 95.03%,96.37%, 96.31%, and 0.9637, respectively.

The results are highly promising. However, our obtained features were over 4000, and we needed to diminish the number of features to an acceptable size in order to have confident results. Therefore, 4096 features were reduced to 250 features using the mRMR method. Then, the reduced features were also classified with classical ML techniques and thus, we created new advanced hybrid algorithms. All these implementations were exhibited in Figure 2, briefly. Figure 2 includes the question “Is the accuracy acceptable?”, which is based on the performance criteria, such as an accuracy above 95%. Table 5 shows performance metrics of advanced hybrid algorithms after using mRMR.

According to Table 5, all advanced hybrid algorithms achieve better performance when compared with Table 4 results. Reduced features using mRMR achieved more effective results.

According to Table 5, we can state that the OzNet-mRMR-kNN and OzNet-mRMR-NB hybrid algorithms hit the top with the same accuracy of 98.42%. However, their other performance metrics are not the same. When OzNet-mRMR-kNN was examined, it had a sensitivity of 98.6%, a specificity of 98.25%, a precision of 98.25%, an F1 score of 98.42%, and an AUC value of 0.9842. When OzNet-mRMR-NB was analyzed, it achieved a sensitivity of 97.54%, a specificity of 99.3%, a precision of 99.29%, an F1 score of 98.41%, and an AUC value of 0.9909. The hybrid algorithm was also successful, and the AUC value was close to 1. Therefore, we could indicate that the best algorithm is OzNet-mRMR-NB.

In conclusion, all these stages indicate that deep learning algorithms achieve reliable results when they are supported with a feature extraction and machine learning algorithm.

Figure 3 shows the confusion matrices of OzNet hybrid algorithms belonging to brain stroke CT images. In addition, Figure 4 shows the ROC curve of the OzNet hybrid algorithms. Figure 5 also indicates a comparison of the hybrid algorithms performance.

## 5. Discussion

In this paper, we designed a novel structure for determining stroke from brain CT images. Essentially, this study has some advantages and disadvantages. First, we presented advantages of the study as follows. (i) We designed a new deep learning algorithm, OzNet. (ii) OzNet is compared with famous algorithms: GoogleNet, Inceptionv3, and MobileNetv2. (iii) A 10-fold cross validation is utilized to get reliable results while training these algorithms, and the classification results showed that OzNet is successful to detect stroke. (iv) Then, 4096 features were extracted from a fully connected layer of OzNet for each image. In this stage, OzNet was employed as a deep feature extractor. (v) The dimension of the extracted features was reduced using mRMR dimension reduction method; hence, 250 features were obtained, and the model reliability was achieved. (vi) Then, 250 features were utilized with classical ML algorithms and tested for the prediction. Therefore, OzNet-mRMR-NB achieved the best performance among the hybrid models, and Table 5 exhibits the performance of hybrid algorithms. The disadvantages of the study are as follows. (i) The binary classes (normal and stroke) were analyzed. We could not use different stroke types. (ii) The CT images were used instead of MRI. Although CT images sometimes have the advantage of accelerating the diagnosis process, MRI is confident. However, stroke needs speedy recognition for patients. Therefore, in this paper, we examined CT images to detect stroke. Additionally, when we evaluate OzNet in signals and images, the results were also very good, detailed in Ozaltin and Yeniay [60] and Ozaltin et al. [7] studies. Further, Table 6 demonstrated results for the comparison with prior works.

When we investigated Table 6, we compared our study with previous studies for detecting stroke. According to Table 6, the proposed hybrid algorithm obtained better performance than the previous studies. The proposed framework achieved better results than the previous studies since we used not only deep learning or machine learning algorithms, but also combined them with a feature reduction method.

## 6. Conclusions

This paper aimed to classify brain stroke CT images using OzNet and hybrid algorithms. The dataset consists of 1900 images of stroke and normal classes. In the formation of hybrid algorithms, OzNet was combined with the feature selection method, mRMR, and ML algorithms in order to improve the classification performance and obtain reliable results. First, the OzNet classification performance was compared with GoogleNet, Incep-tionresnetv3, and MobileNetv2. In this stage, though it achieved the highest accuracy of 87.47%, we continued to analyze and needed to obtain reliable results of the investigated vital disease. Then, we utilized OzNet as a deep feature extractor from images, and its last layer was applied to get 4096 features. Second, 4096 features were classified with classical ML algorithms, but the obtained features were over 4000, and the number of features should be reduced to obtain reliable results. Therefore, we reduced the dimension of features utilizing the mRMR method. This method not only gives the best relationship features, but it also examines minimum residuals from each other. Next, the reduced features were classified using classical ML algorithms. As a result, OzNet-mRMR-NB that created the new structure achieved an accuracy of 98.42% to detect stroke from brain CT images.

A stroke is a feared neurological problem in the world because it may cause death or physical disability. Therefore, a rapid diagnosis is crucial for patients and clinicians. Artificial intelligence is in every aspect of our lives. Using it to detect these and similar problems in the field of health will speed up the process considerably. But it is very important to obtain safe results. Although we presented a kind of artificial intelligence algorithm in this study, we increased the reliability of the results with ML and feature selection algorithms. In future studies, auxiliary support algorithms will be created that increase the reliability of the results by taking advantage of the power of artificial intelligence.

## Figures and Tables

**Figure 1 bioengineering-09-00783-f001:**
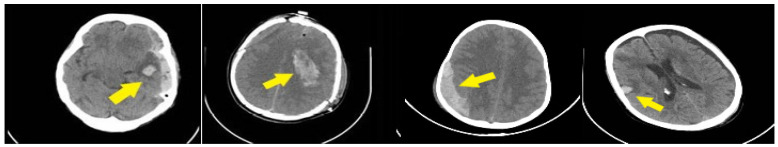
Stroke instances from the dataset.

**Figure 2 bioengineering-09-00783-f002:**
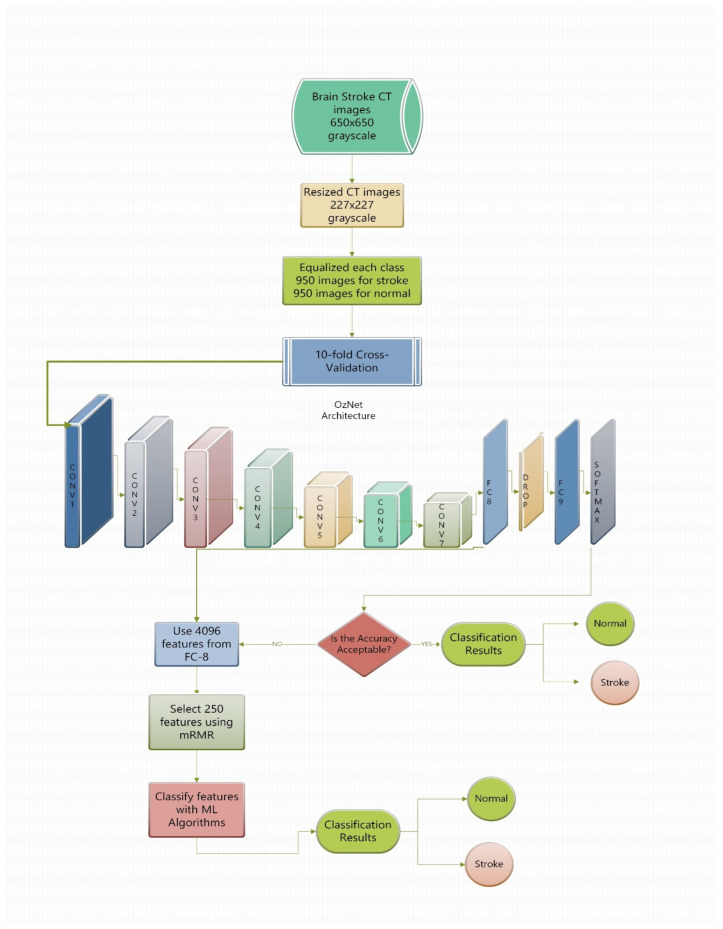
Overview of the proposed framework.

**Figure 3 bioengineering-09-00783-f003:**
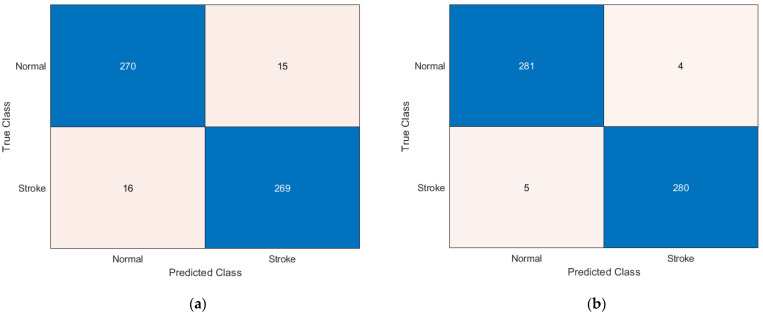
Confusion Matrices of OzNet hybrid algorithms (**a**) OzNet-mRMR-DT. (**b**) OzNet-mRMR-kNN. (**c**) OzNet-mRMR-LDA. (**d**) OzNet-mRMR-NB. (**e**) OzNet-mRMR-SVM.

**Figure 4 bioengineering-09-00783-f004:**
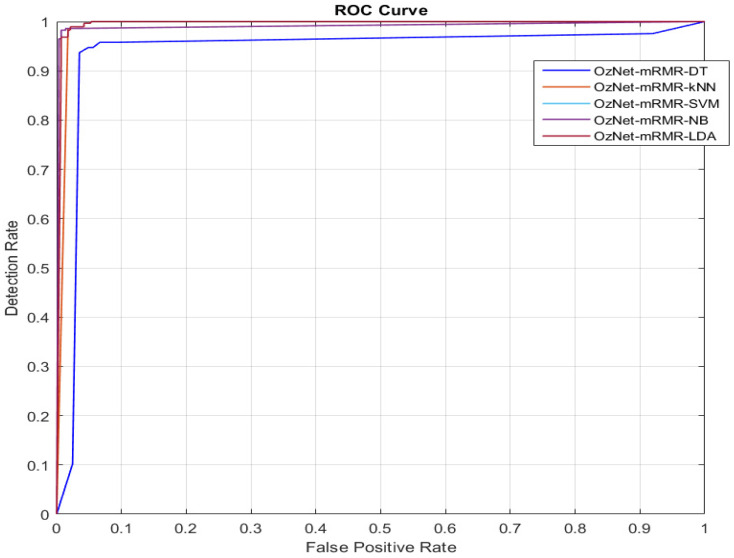
ROC Curves of OzNet hybrid algorithms.

**Figure 5 bioengineering-09-00783-f005:**
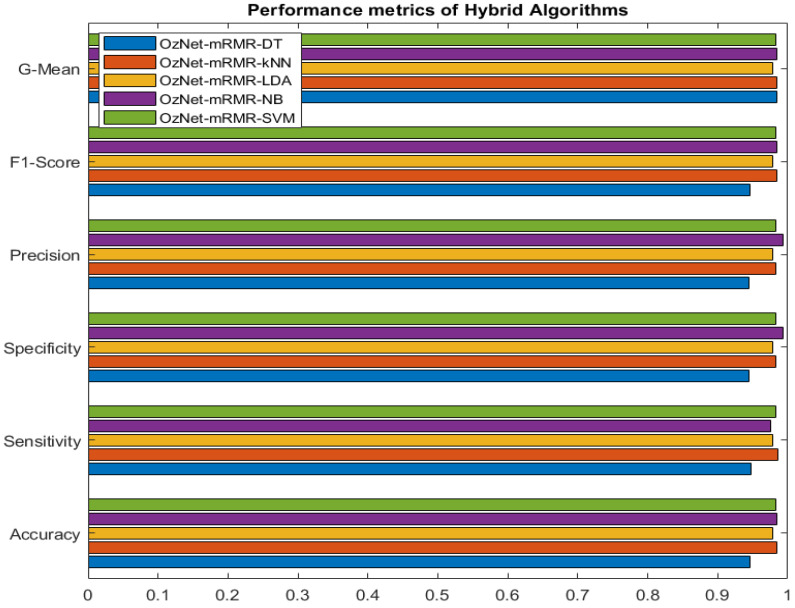
Performance metrics graph of different algorithms.

**Table 1 bioengineering-09-00783-t001:** Parameters details of OzNet.

Layer Name	Type	Size	Filters	Stride	Padding	Output
Input	Input					227 × 227 × 3
Conv-1	Convolution 2D	64	5 × 5	1	1	225 × 225 × 64
MaxPool-1	Max Pooling		3 × 3	2	0	112 × 112 × 64
Conv-2	Convolution 2D	128	3 × 3	1	1	112 × 112 × 128
MaxPool-2	Max Pooling		3 × 3	2	0	55 × 55 × 128
Conv-3	Convolution 2D	128	13 × 13	1	0	55 × 55 × 128
MaxPool-3	Max Pooling		3 × 3	2	0	27 × 27 × 128
Conv-4	Convolution 2D	256	7 × 7	1	1	27 × 27 × 256
MaxPool-4	Max Pooling		2 × 2	2	0	13 × 13 × 256
Conv-5	Convolution 2D	128	3 × 3	1	1	13 × 13 × 128
MaxPool-5	Max Pooling		3 × 3	2	0	6 × 6 × 128
Conv-6	Convolution 2D	128	3 × 3	1	1	6 × 6 × 128
MaxPool-6	Max Pooling		3 × 3	2	0	3 × 3 × 128
Conv-7	Convolution 2D	128	3 × 3	1	1	3 × 3 × 128
MaxPool-7	Max Pooling	-	2 × 2	2	0	1 × 1 × 128
FC-8	Fully Connected	4096				1 × 1 × 4096
Drop-8	Dropout	50%				
FC-9	Fully Connected	number of classes				1 × 1 × (number of classes)
Softmax	Softmax					1 × 1 × (number of classes)
Output	Classification	entropy				

**Table 2 bioengineering-09-00783-t002:** Equations for performance metrics.

Performance Metrics	Equations
Accuracy	$\left(TP+TN\right)/\left(TP+TN+FP+FN\right)$
F1-Score	$\left(2\times TP\right)/\left(2\times TP+FP+FN\right)$
G-Mean	$\sqrt{Sensitivity\;\times Specificity}$
Precision	$TP/\left(TP+FP\right)$
Sensitivity	$TP/\left(TP+FN\right)$
Specificity	$TN/\left(TN+FP\right)$

**Table 3 bioengineering-09-00783-t003:** Performance metrics of different architectures on Brain Stroke CT images.

Architectures	Performance Metrics						
	Sensitivity	Specificity	Precision	F1-Score	G-Mean	Accuracy	AUC
GoogleNet	0.8095	0.7789	0.7855	0.7973	0.7941	0.7942	0.8761
Inceptionv3	0.7526	0.7621	0.7598	0.7562	0.7574	0.7574	0.8399
MobileNetv2	0.8768	0.8705	0.8713	0.8740	0.8736	0.8736	0.9407
OzNet	0.8716	0.8779	0.8771	0.8743	0.8747	0.8747	0.9488

**Table 4 bioengineering-09-00783-t004:** Performance metrics of algorithms on Brain Stroke CT images.

Algorithms	Performance Metrics						
	Sensitivity	Specificity	Precision	F1-Score	G-Mean	Accuracy	AUC
OzNet	0.8716	0.8779	0.8771	0.8743	0.8747	0.8747	0.9488
OzNet-DT	0.9023	0.9053	0.9050	0.9036	0.9038	0.9038	0.9238
OzNet-kNN	0.9489	0.9579	0.9575	0.9532	0.9534	0.9534	0.9534
OzNet-LDA	0.9669	0.9609	0.9611	0.9640	0.9639	0.9639	0.9937
OzNet-NB	0.9774	0.9489	0.9503	0.9637	0.9631	0.9632	0.9674
OzNet-SVM	0.9594	0.9714	0.9711	0.9652	0.9654	0.9654	0.9921

**Table 5 bioengineering-09-00783-t005:** Performance metrics of advanced hybrid algorithms on Brain Stroke CT images.

Algorithms	Performance Metrics						
	Sensitivity	Specificity	Precision	F1-Score	G-Mean	Accuracy	AUC
OzNet-mRMR-DT	0.9474	0.9439	0.9441	0.9457	0.9456	0.9456	0.9403
OzNet-mRMR-kNN	0.9860	0.9825	0.9825	0.9842	0.9842	0.9842	0.9842
OzNet-mRMR-LDA	0.9789	0.9789	0.9789	0.9789	0.9789	0.9789	0.9984
OzNet-mRMR-NB	0.9754	0.9930	0.9929	0.9841	0.9842	0.9842	0.9909
OzNet-mRMR-SVM	0.9825	0.9825	0.9825	0.9825	0.9825	0.9825	0.9956

**Table 6 bioengineering-09-00783-t006:** Performance comparison with prior works.

Works	Data Type	Classifier	Accuracy
Chin et al. [26]	Brain CT	CNN	90%
Shalikar et al. [61]	Brain CT	SVM	90%
Marbun et al. [62]	Brain CT	CNN	90%
Diker et al. [63]	Brain CT	VGG-19	97.06%
Raghavendra et al. [64]	Brain CT	PNN (Probabilistic Neural Network)	94.37%
This study	Brain CT	OzNet-mRMR-NB	98.42%

## Data Availability

https://www.kaggle.com/datasets/afridirahman/brain-stroke-ct-image-dataset [accessed on 1 December 2021].

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
