# Peer review of "A Deep Learning Approach for Detecting Stroke from Brain CT Images Using OzNet"

_bioengineering, 2022, doi:10.3390/bioengineering9120783_

Round 1

Reviewer 1 Report

The paper proposes an automatic method for stroke detection in CT images based on proposed new deep network structure – OzNet. Several issues should be addressed in the revised version of this paper:

1.     The literature review does not come to a conclusion. It should show what are the shortcomings of current methods of detecting strokes in CT images and to what extent the proposed method will remove the limitations of existing methods.

2.     Then, a novelty of this work should be clearly specified.

3.     Please demonstrate the advantages of OzNet when compared to other CNN architectures. Also, numerical experiment will be needed to compare obtained results with results provide by classic CNNs (for the same dataset)

4.     The idea of combining the OzNet and classic classifier is not clear (Fig. 2). What “Enough Accuracy” means? Please explain this idea in more details.

5.     Fig. 2 suggests that there are over 4000 features fed to the classifier input. Is it correct? If so, the feature reduction technique is needed to limit the number of features to reasonable size. Otherwise obtained classification results are not reliable. The input data space is very sparse because the number of dimensions is very large when compared to the number of samples. Thus, it not possible to obtain any generalization of the classifier.

6.     How were the classifiers validated?  

7.     Discussion should be expanded. Analysed datasets should be defined when comparison with other researchers is performed. What is a reason to present results for MR images? The authors should also explain, why their approach outperforms others.

Author Response

The manuscript has been revised in accordance with the comments of the reviewers. Reviewer’s comments have been carefully considered and all revisions recommended by the reviewers have been incorporated. Based on the discussion above, I hope you will re-evaluate and send back to reviewers. I hope you and reviewers will be satisfied from this revision and you will reconsider it for publication. All corrections were highlighted in the article by using red and yellow font color.

Sincerely Yours,

Oznur Ozaltin

Reviewer 2 Report

This article is generally interesting and well written. My minor comments aim to increase the scientific soundness and clarity of it.

Line 7-8  – Please correct affiliation (city, country).

Line 51 – Please use some arrows (or colour lines) to precisely point areas affected by the strokes in CT images of figure 1. By the way, what is an idea of figure 1?

Line 54-58 – Please explain your hypothesis properly.

Line 63 – Please explain briefly what are normal and abnormal stroke classes. Could stroke even be “normal” ?

Line 72 – Why the authors introduced PACS, LACS and TACS abbreviations if they are not further used ? The same is true in relation to many more acronyms like FODPSO, FCN, HIMS, DICOM etc.

Line 97 – K-Nearest Neighbors is either abbreviated to KNN or to kNN (line 213). Please select one.

Line 159 – CT was already abbreviated in line 37.

Line 356 – In the present form the conclusion is just repetition of the methods and results. This must be corrected. Some future perspectives are also needed.

Author Response

(The authors gave the same response as above.)

Round 2

Reviewer 1 Report

Thank you for considering my comments, I am mostly satisfied with the answers. A few minor issues remain:

1. "good enough" is used in the introduction, but is not the right term. It needs to be replaced by a more quantitative statement.

2. "the studies in the literature were examined" is repeated twice in the text

3. OzNet is not a new architecture as it has already been published in [7]

4. "Acceptable" for whom? Please explain

5. 250 features is still a lot. In future works, I recommend further reduction of the feature number. See e.g.

https://doi.org/10.1007/978-3-031-09135-3_15

(include this work in the literature review) where the discrimination between stroke and brain tumor was made with much fewer features and classic ML algorithms.

6. Provide a reference to the mRMR feature reduction method
